# Tumor-Suppressive Effect of Metformin via the Regulation of M2 Macrophages and Myeloid-Derived Suppressor Cells in the Tumor Microenvironment of Colorectal Cancer

**DOI:** 10.3390/cancers14122881

**Published:** 2022-06-10

**Authors:** Joyeon Kang, Doyeon Lee, Kyoung Jin Lee, Jaepil Eric Yoon, Ji-Hee Kwon, Yoojeong Seo, Janghyun Kim, Shin Young Chang, Jihye Park, Eun Ae Kang, Soo Jung Park, Jae Jun Park, Jae Hee Cheon, Tae Il Kim

**Affiliations:** 1Institute of Gastroenterology, Yonsei University College of Medicine, Seoul 03722, Korea; jykang@eutilex.com (J.K.); leedoyeon1@hanmail.net (D.L.); minishell87@naver.com (K.J.L.); ericyoon93@gmail.com (J.E.Y.); fulljh@hanmail.net (J.-H.K.); yjseo90@yuhs.ac (Y.S.); kphe22@gmail.com (J.K.); shinyoungchan112@yuhs.ac (S.Y.C.); wisdompark@yuhs.ac (J.P.); eakang@yuhs.ac (E.A.K.); sjpark@yuhs.ac (S.J.P.); jaejpark@yuhs.ac (J.J.P.); geniushee@yuhs.ac (J.H.C.); 2Graduate School of Medical Science, Brain Korea 21 Project, Yonsei University College of Medicine, Seoul 03722, Korea; 3Department of Internal Medicine, Yonsei University College of Medicine, Seoul 03722, Korea; 4Cancer Prevention Center, Yonsei University College of Medicine, Seoul 03722, Korea

**Keywords:** metformin, MDSC, M2 macrophage, tumor microenvironment, colorectal cancer

## Abstract

**Simple Summary:**

The tumor suppressing effect of metformin has been reported, and tumor microenvironment including immune cells contribute to tumor progression in colorectal cancer. However, the effects of metformin on tumor-promoting MDSCs and M2 macrophages and its mechanisms remain unclarified. Here, we demonstrated that metformin-induced activation of AMPK and subsequent mTOR inhibition decreased the MDSC and M2 macrophage fractions by downregulating the mevalonate pathway. Metformin may be a valuable drug for potential CRC prevention and treatment strategies by regulating the immune cells of the tumor microenvironment and tumor cells.

**Abstract:**

Myeloid-derived suppressor cells (MDSCs) and M2 macrophages in the tumor microenvironment contribute to tumor progression by inducing immune tolerance to tumor antigens and cancer cells. Metformin, one of the most common diabetes drugs, has shown anti-inflammatory and anti-tumor effects. However, the effects of metformin on inflammatory cells of the tumor microenvironment and its underlying mechanisms remain unclarified. In this study, we investigated the effect of metformin on M2 macrophages and MDSCs using monocyte THP-1 cells and a dextran sodium sulfate (DSS)-treated Apc^Min/+^ mouse model of colon cancer. Metformin decreased the fractions of MDSCs expressing CD33 and arginase, as well as M2 macrophages expressing CD206 and CD163. The inhibitory effect of metformin and rapamycin on MDSCs and M2 macrophages was reversed by the co-treatment of Compound C (an AMP-activated protein kinase (AMPK) inhibitor) or mevalonate. To examine the effect of protein prenylation and cholesterol synthesis (the final steps of the mevalonate pathway) on the MDSC and M2 macrophage populations, we used respective inhibitors (YM53601; SQLE inhibitor, FTI-277; farnesyl transferase inhibitor, GGTI-298; geranylgeranyl transferase inhibitor) and found that the MDSC and M2 populations were suppressed by the protein prenylation inhibitors. In the DSS-treated Apc^Min/+^ mouse colon cancer model, metformin reduced the number and volume of colorectal tumors with decreased populations of MDSCs and M2 macrophages in the tumor microenvironment. In conclusion, the inhibitory effect of metformin on MDSCs and M2 macrophages in the tumor microenvironment of colon cancers is mediated by AMPK activation and subsequent mTOR inhibition, leading to the downregulation of the mevalonate pathway.

## 1. Introduction

Colorectal cancer (CRC) is the third most common cancer worldwide and is a complex tumor condition caused by various genetic and environmental factors. Recent advancements in cancer screening and treatment have improved the prevention and outcomes of CRC [1]. However, the scientific understanding of CRC initiation and progression remains limited and elusive in various aspects. One flourishing field in cancer treatment is a therapeutic strategy targeting the tumor microenvironment, including immune cells, and the heterogeneous and intricate microenvironment which promotes tumor growth [2].

The tumor microenvironment comprises blood vessels, fibroblasts, inflammatory cells, and the extracellular matrix [3]. In this heterogeneous microenvironment, macrophages and myeloid progenitors play crucial roles in cancer regression and survival [2,3]. Macrophages exhibit both pro- and anti-tumor functions. Termed M1 and M2 polarized macrophages, the two subsets display different phenotypes with differing functions. The M1 phenotype is activated by Toll-like receptor ligands, and interferon gamma (IFN-γ), produces tumor cytotoxicity by producing pro-inflammatory cytokines such as TNF-α and IL-1β, and inhibits tumor progression. In contrast, the alternatively activated M2 phenotype is activated by cytokines IL-4 and IL-13. M2 macrophages promote and produce high levels of cytokines that stimulate tumor growth and progression via the production of anti-inflammatory stimuli such as TGF-β1 and IL-10 [4,5,6]. Thus, macrophages both positively and negatively influence tumor growth [4]. In addition, myeloid-derived suppressor cells (MDSCs) are associated with greater tumor burdens and worse prognosis [7]. MDSCs are actively recruited to primary and metastatic tumor sites. This process is regulated by chemokines produced by the tumor, with little specificity in the chemokine type. Inflammatory factors that induce MDSC recruitment and expansion in the tumor microenvironment, such as IL-6, IL-10, and IL-1β, display potent immunosuppressive and tumor-promoting functions in the tumor microenvironment via multiple mechanisms, such as immunosuppressive cell induction, lymphocyte homing blockade and reactive oxygen and nitrogen species production [8]. Amid the complex dichotomy of M1 and MDSC/M2 leukocytes, few studies have explored the effects of therapeutic drugs on these populations.

Metformin, a biguanide subclass molecule, is the most commonly prescribed drug in type II diabetes treatment. This drug reduces hepatic gluconeogenesis and insulin resistance by increasing peripheral glucose uptake and consumption in the liver and skeletal muscles by inhibiting the oxidative phosphorylation pathway. In addition, as a direct anti-tumor effect of metformin, AMP-activated protein kinase (AMPK) activation-mediated mTOR inhibition, the suppression of cancer stem cells, and the inhibition of cellular transformation are involved in the mechanisms of tumor suppression by metformin [9,10]. Moreover, regarding the anti-tumor effect, metformin targets the respiratory chain complex I in the mitochondria, leading to altered metabolism, including inhibition of the tricarboxylic acid (TCA) cycle and oxidative phosphorylation (OXPHOS) [9,10]. Clinically, metformin also reduces the development and recurrence of colorectal polyps, increases CRC survival in DM patients and decreases the recurrence of polyps in non-diabetic patients [11]. Therefore, metformin was recently deemed a potential adjunctive drug in cancer treatment or a chemopreventive agent due to its anti-tumor capabilities and relative safety in the human body. In addition, recent studies have shown that metformin exhibits anti-inflammatory effects in human vascular endothelial cells and smooth muscle cells via the AMPK pathway [12]. These findings suggest that the impact of metformin on tumors may be related to both a direct effect on tumor cells and an inflammatory cell-mediated effect on the tumor microenvironment [13]. Therefore, we investigated the effect of metformin on tumor-promoting MDSCs and M2 macrophages and its mechanism in the tumor microenvironment of CRC.

The present study demonstrated that metformin-induced activation of AMPK decreased the MDSC and M2 macrophage fractions by downregulating the mevalonate pathway. Metformin may be a valuable drug for potential CRC prevention and treatment strategies by regulating the immune cells of the tumor microenvironment and tumor cells.

## 2. Materials and Methods

### 2.1. Cell Culture and Reagents

The human monocytic myeloid cell line THP-1 (KCLB, Seoul, Korea) was derived from an acute monocytic leukemia patient. THP-1 cells were cultured in RPMI-1640 (Gibco-Life Technologies, Grand Island, NY, USA) supplemented with 10% fetal bovine serum (Gibco-Life Technologies, Grand Island, NY, USA) and 1% penicillin/streptomycin (Invitrogen, Carlsbad, CA, USA) at 37 °C in 5% CO_2_. THP-1 cells were seeded in the medium and cultured with or without 0.25 to 5 mM metformin and 100 nM PMA (Sigma, St. Louis, MO, USA), 200 μM mevalonate (Santa Cruz Biotechnology, Santa Cruz, CA, USA), 50 and 125 μM AICAR, 20 μM Compound C, 20 and 50 nM rapamycin and 2 μM simvastatin (Merck Millipore, Darmstadt, Germany). In addition, 10 and 20 μM FTI-277 (Sigma, St. Louis, MO, USA) and 5 and 10 μM GGTI-298 (Tocris Bioscience, Bristol, UK) and YM-53601 (Cayman Chemical, Ann Arbor, MI, USA) were used to treat THP-1 cells.

### 2.2. Flow Cytometry Analysis of M2 Macrophages and MDSCs

Before flow cytometry analysis, metformin, mevalonate and other reagents were used to treat THP-1 cells plated at a density of 2 × 10^6^ in six-well plates in 2 mL of the medium and incubated at 37 °C in 5% CO_2_ for 48 h. FACS buffer (1× phosphate-buffered saline (PBS), 1% bovine serum albumin and 2 mM ethylene diamine tetra-acetic acid) was used in the flow cytometry analysis of macrophage marker antibodies (PE-Cy3-conjugated anti-CD68), MDSC marker antibodies (FITC-conjugated anti-CD33 and anti-Arginase-1), M2 macrophage marker antibodies (FITC-conjugated anti-CD206 and anti-CD163) and M1 macrophage marker antibodies (FITC-conjugated anti-iNOS), which have been known to be used as markers for each cell population [14]. Primary antibodies (Santa Cruz Biotechnology, Santa Cruz, CA, USA) and secondary antibodies (FITC-rabbit, goat, and Cy3-mouse; Abmgood, Vancouver, Canada) were added and incubated for 10 min at 4 °C. The cells were washed with FACS buffer and analyzed using a FACSVerse instrument (BD Biosciences, San Diego, CA, USA) coupled with a computer with BD FACSuite software for data analysis.

### 2.3. Western Blotting

To analyze the AMPK-mTOR pathway and HMG-CoA reductase (HMGCR) of mevalonate pathway, the expressions of phosphorylated AMPK (p-AMPK), phosphorylated S6 (pS6) and HMGCR were evaluated. The cells were treated with metformin (1 mM, 2 mM, or 5 mM) in 2 mL of medium in six-well plates, and incubated for 48 h. For HMGCR, the cells were treated with100 μM PMA, 5 mM metformin, 50 nM rapamycin and 0.5 mM AICAR for 24 h. All cells were harvested and washed twice with PBS, pelleted by centrifugation, and lysed at 4 °C for 15 min in protein extraction solution (iNtRON Biotechnology, Gyeonggi-do, South Korea). The protein concentrations of the samples were determined using the bicinchoninic acid protein assay (Thermo Fisher Scientific, Rockford, IL, USA). Next, the samples were subjected to sodium dodecyl sulfate-polyacrylamide gel electrophoresis, followed by blotting onto polyvinylidene fluoride membranes (Bio-Rad, Hercules, CA, USA) in a methanol-based tris-glycine buffer. The membranes were then blocked with 5% skim milk/TBST (Tris-buffered saline and Tween 20) for 1 h. Each of the primary antibodies (anti-AMPK, anti-p-AMPK (Thr172), anti-S6, anti-p-S6 (Ser235/236); Cell Signaling Technology, Danvers, MA, USA, anti-HMGCR; abcam, Cambridge, MA, USA) were incubated with the membranes with continuous mixing overnight at 4 °C. The membranes were rinsed at least three times with TBST and then incubated with secondary antibodies for 1 h at room temperature. After the final TBST rinse, the enhanced chemiluminescence (ECL) Western blotting detection kit (Amersham Biosciences, Freiburg, Germany) was used, followed by exposure of the membranes to Kodak film to express the emission of proteins.

### 2.4. ELISA

To measure PGE_2_, the cells were plated at a density of 1 × 10^6^ cells/well in six-well plates, co-treated with PMA for macrophage differentiation, and treated with metformin and/or mevalonate for 48 h. The levels of PGE_2_ in the supernatants were determined using the PGE_2_ ELISA assay kit (ADI-900-001; Enzo Life Sciences, Farmingdale, NY, USA) according to the manufacturer’s protocol. Enzymatic reactions were measured by spectrophotometry at 405 nm.

### 2.5. In Vivo Experiments Using a Mouse Model of Colorectal Cancer

Six-week-old male C57BL/6J Apc^Min/+^ mice (Jackson Laboratory, Bar Harbor, ME, USA) were used. The mouse experiment was approved by the Ethics Committee, IACUC of YUHS (Institutional Animal Care and Use Committee Yonsei University Health System, Approval Code: 2017-0328), and performed according to the institutional guidelines and policies. All of the mice were subjected to 3% DSS for 6 days daily via drinking water and were allowed to recover by drinking regular water for 3 weeks. The experimental group was injected with 250 or 350 mg/kg of metformin i.p. (intraperitoneal injection), while the control group was injected with PBS. After 3 weeks, all of the mice were sacrificed and Swiss rolls of the large intestines were fixed in 4% paraformaldehyde (PFA) and embedded in paraffin blocks for immunohistochemistry staining.

### 2.6. Immunohistochemical Staining

Paraffin-embedded sections were de-paraffinized in xylene and rehydrated in gradually decreasing concentrations of ethanol. Antigen retrieval was performed using a sodium citrate buffer (10 mM, pH 6.0) in a heated pressure cooker for 5 min. After incubation with 3% hydrogen peroxide to block endogenous peroxidase activity for 30 min, the sections were incubated in 5% BSA/TBS solution for 30 min at room temperature. Anti-CD11b (MDSC marker; 1:4000 dilution; Abcam, MA, USA), anti-CD206 (M2 marker; 1:200 dilution; Abcam, MA, USA) and anti-CD86 (M1 marker; 1:200 dilution; Santa Cruz Biotechnology, Santa Cruz, CA, USA) were incubated with the sections overnight at 4 °C, and then secondary antibodies were incubated for 30 min at room temperature. After the slides were detected using a Vectastain ABC kit (Vector Laboratories, Burlingame, CA, USA), immunostaining was performed using a DAB solution (Dako, Carpinteria, CA, USA). After counterstaining with hematoxylin, IHC staining was evaluated by light microscopy and immunoactivity was assessed based on the proportion of immunostained MDSCs and M2 and M1 macrophages counted in ten different fields for three samples with 200× magnification.

### 2.7. Real-Time PCR

The total RNA of THP-1 cells was isolated using TRIzol reagent (Gibco-Life Technologies, Grand Island, NY, USA). cDNA synthesis from 2 μg of total RNA was performed using Reverse Transcription Master Premix (ELPISBIOTECH, Daejeon, Korea). Real-time qPCR was performed using SYBR Green Master mix (Enzynomics, Daejeon, Korea) and the following primers: HMGCR (forward, 5′-CCCAGCCTACAAGTTGGAAA-3′; reverse, 5′-AACAAGCTCCCATCACCAAG-3′), ITGAM (forward, 5′-GAGCAGGGGTCATTCGCTAC-3′; reverse, 5′-GCTGGC TTA GATGCGATGGT-3′), Mrc1 (forward, 5′-AGGGACCTGGATGGATGACA-3′; reverse, 5′-TGTACCGCACCCTCCATCTA-3′), and GAPDH and β-actin as a housekeeping gene.

### 2.8. Statistical Analysis

Statistical analysis was performed using IBM SPSS Statistics version 20.0 (IBM Co., Armonk, NY, USA). Student’s two-tailed *t*-test was performed to analyze differences between experimental groups. *p* values lower than 0.05 were considered statistically significant.

## 3. Results

### 3.1. Metformin, an AMPK Regulator, and Mevalonate Modulate the MDSC and M2 Macrophage Fractions in THP-1 Cells

Metformin, a drug widely used to treat type 2 diabetes, activates AMPK. First, we performed flow cytometry to assess the proportions of MDSCs (CD33 and Arg-1 markers) and M2 macrophages (CD206 and CD163 markers) after treatment with 0.25 mM to 2 mM metformin for 48 h. The proportions of MDSCs and M2 macrophages decreased significantly after metformin treatment in a dose-dependent manner (Figure 1A and Appendix A), while the M1 macrophage proportion did not change with metformin treatment (Appendix A). We further examined whether the AMPK pathway was involved in reducing the MDSC and M2 macrophage populations using AICAR (AMPK activator), Compound C (AMPK inhibitor) and metformin. Both metformin and AICAR reduced the populations of MDSCs and M2 macrophages in a similar fashion. However, Compound C increased the proportions of MDSCs and M2 macrophages (Figure 1B). In addition, co-treatment of Compound C and metformin reversed the inhibitory effects of metformin on MDSCs and M2 macrophages, suggesting that the inhibitory effect of metformin on MDSCs and M2 macrophages occurred by activating AMPK (Figure 1B and Appendix A). To confirm the signals using the AMPK-mTOR signaling pathway, we examined the activation of p-AMPK and p-S6 using Western blotting. As expected, treatment with metformin, AICAR and rapamycin (mTOR inhibitor) increased AMPK activation and decreased p-S6 (Figure 1C).

We previously showed that metformin suppresses cancer stem cells via AMPK activation and mevalonate pathway inhibition in colon cancer cells [15]. We sought to investigate whether the mevalonate pathway also plays a role in the progression of MDSCs and M2 macrophages in the tumor microenvironment. Mevalonate increased the fractions of MDSCs and M2 macrophages compared to the control. Co-treatment of metformin and mevalonate reversed the inhibitory effects of metformin on MDSCs and M2 macrophages compared to the metformin treatment group (Figure 1D and Appendix A). These results indicate that AMPK activation by metformin regulates the proportions of MDSCs and M2 macrophages, and this effect is related to the mevalonate pathway.

### 3.2. Rapamycin and Simvastatin Reduce the MDSC and M2 Macrophage Fractions, an Effect That Is Reversed by Mevalonate Treatment

Simvastatin is an HMG-CoA reductase inhibitor that primarily inhibits the mevalonate pathway and is a cholesterol-lowering agent [16]. To further confirm the relationship between the mevalonate pathway and AMPK/mTOR pathway, THP-1 cells were treated with rapamycin and simvastatin with or without mevalonate, and flow cytometry was performed. First, rapamycin (mTOR inhibitor) treatment induced the same inhibitory effect on the proportions of MDSCs and M2 macrophages and mevalonate treatment reversed the inhibitory effects of rapamycin on MDSCs and M2 macrophages (Figure 2A and Appendix A). Next, to confirm the signals through the mevalonate pathway, THP-1 cells were treated with simvastatin with or without mevalonate and flow cytometry was performed. Inhibition of HMG-CoA reductase by simvastatin reduced the proportions of MDSCs and M2 macrophages derived from THP-1 cells. In addition, when THP-1 cells were treated with a combination of simvastatin and mevalonate, the reduced MDSC and M2 macrophage fractions by simvastatin were reversed by mevalonate (Figure 2B and Appendix A). Furthermore, to show the same effect of metformin in the macrophage-activated state, we induced macrophage activation using PMA and conditional media (CM) of 18Co cells (intestinal myofibroblasts), critical cell components of the tumor microenvironment [14]. Flow cytometry analysis revealed that treatment with PMA and 18Co CM increased the MDSC/M2 macrophage population derived from THP-1 cells, and this induced population was reduced by treatment with metformin and rapamycin (Figure 2C,D and Appendix A). We confirmed that the mevalonate and AMPK/mTOR pathways regulate the MDSC and M2 macrophage populations.

### 3.3. Tumor-Suppressive Effects of Metformin in a Mouse Colon Cancer Model

To prove the effect of metformin in an in vivo mouse tumorigenesis model, colon tumors were developed using Apc^Min/+^ mice with dextran sodium sulfate (DSS) treatment (Figure 3A). In this mouse colon cancer model, treatment with metformin significantly decreased the number and size of colonic tumors compared to those of the control group (Figure 3B,C). Regarding immune cells in the tumor microenvironment, immunohistochemistry staining of MDSC (CD11b), M2 (CD206) and M1 (CD86) markers in mouse colon tumors showed that the numbers of MDSC- and M2-positive cells decreased remarkably in the metformin-treated mice compared to those in the control mice (Figure 3D). In addition, CD11b and CD206 mRNA levels in the metformin-treated mouse colon tumors were also significantly reduced compared to those in the control mice (Figure 3E). However, M1-positive cells showed no significant changes (Appendix A). We confirmed the in vitro study findings in vivo, revealing that metformin decreased the MDSC and M2 macrophage populations. Taken together, these results indicate that metformin regulates the pro-tumor immune cell populations in the microenvironment.

### 3.4. Metformin Decreases the MDSC and M2 Macrophage Populations by Downregulating the Mevalonate Pathway

HMG-CoA reductase (HMGCR) is the rate-controlling enzyme of the mevalonate pathway and converts HMG-CoA to mevalonate [17,18]. Metformin is a potent HMGCR inhibitor 16, and mTOR, which is inhibited by AMPK activation, also targets HMGCR [19]. Therefore, we analyzed the expression level of *HMGCR* after metformin treatment or mTOR inhibition. *HMGCR* expression in THP-1 cells was increased when treated with PMA only, whereas PMA-induced *HMGCR* expression was attenuated by treatment with metformin, rapamycin (Figure 4A), and AICAR (Figure 4B). At the same time, in Western blotting of HMGCR, we found a significant decrease in the HMGCR protein expression caused by the treatment with metformin, rapamycin, and AICAR (Figure 4C). Thus, metformin activates AMPK, downregulating HMGCR and decreasing the proportion of MDSCs/M2 macrophages by downregulating the mevalonate pathway.

The mevalonate pathway is responsible for the synthesis of cholesterol and protein prenylation, which includes protein farnesylation and geranyl-geranylation [20]. In addition, metformin suppresses cancer stem cells by inhibiting protein prenylation in the mevalonate pathway in colon cancer cells [15]. Therefore, we sought to investigate whether the downregulation of MDSC/M2 macrophage by metformin is attributed to the attenuation of cholesterol synthesis or protein prenylation. THP-1 cells were treated with YM-53601 (cholesterol synthase inhibitor), FTI-277 (farnesyl transferase inhibitor) and GGTI-298 (geranylgeranyl transferase inhibitor), and the proportions of MDSCs/M2 macrophages were analyzed by flow cytometry analysis (Figure 4D and Appendix A). YM-53601, FTI-277 and GGTI-298 treatment alone did not reduce the MDSC/M2 population. However, the combination of FTI-277 and GGTI-298 noticeably suppressed the MDSC/M2 population derived from THP-1 cells (Figure 4D), suggesting that the proportion of MDSC/M2 macrophage cells decreases by downregulating protein prenylation (Appendix A). However, in Western blot analysis performed to identify a change in the shifted prenylated protein band of RAS, despite decreased expression of RAS by metformin treatment, we could not find any definite shifted prenylated protein band of RAS (Appendix A).

Furthermore, the activation of the mevalonate pathway and ERK induces PGE_2_ production [21], which is related to M2 macrophage differentiation [22]. Therefore, we performed the enzyme-linked immunosorbent assay (ELISA) to measure the PGE_2_ concentration of the supernatant of THP-1 cells after treatment with metformin and/or mevalonate co-treated with PMA for 48 h. The PGE_2_ level was increased when the cells were treated with PMA alone, and PMA-induced PGE_2_ was suppressed by metformin co-treatment. In addition, mevalonate induced a further increase in PMA-induced PGE_2_ production, and this mevalonate-induced effect was also attenuated by metformin co-treatment (Figure 4E). Taken together, these results suggest that AMPK activation and subsequent mTOR inhibition by metformin is related to decreased MDSC/M2 macrophage differentiation via the suppression of HMGCR, leading to the inhibition of the mevalonate pathway, possibly attenuating PGE_2_ production (Figure 5).

## 4. Discussion

The tumor microenvironment comprises various components, such as the extracellular matrix, stromal cells (e.g., fibroblasts and myofibroblasts), immune cells, and endothelial cells [3]. In addition to harboring carcinoma cells, a tumor microenvironment comprises various components that play a major role in influencing the outcome of the malignancy [3]. When recruiting the tumor microenvironment, immune cells in particular are deeply involved with the progression and regression of the tumor.

Metformin has an anti-tumor effect in many aspects of tumorigenesis. Metformin leads to cell cycle arrest and reduces cell growth, proliferation, protein synthesis and the cancer stem cell number in the tumor and its microenvironment by activating the AMPK pathway and inhibiting the mTOR pathway or PI3K/AKT pathway [12,13,23]. The present study demonstrated that metformin yields favorable outcomes in the immune cells of the tumor microenvironment. In assembling and generating this body of work, metformin, the mTOR inhibitor, the AMPK activator and the HMG-CoA reductase inhibitor simvastatin suppressed the MDSC/M2 macrophage populations, while mevalonate and the AMPK inhibitor reversed that phenomenon, suggesting that the AMPK/mTOR pathway and mevalonate pathway could be key mediators of MDSC/M2 macrophage regulation.

Moreover, when THP-1 cells were activated to differentiate into macrophages by PMA and 18Co CM, metformin and rapamycin significantly decreased the MDSC/M2 macrophage fraction measured by flow cytometry analysis. In addition, in the tumorigenesis mouse model, metformin-treated mice specifically repressed MDSCs and M2 macrophages without a significant change in the M1 macrophage fraction, suggesting that the effect of metformin is more specific to MDSC/M2 macrophages in the CRC microenvironment.

Interestingly, AMPK activation/mTOR inhibition also inhibits the synthesis of mevalonate and various intermediates of the mevalonate pathway [17]. Mevalonate, an intermediate of the mevalonate pathway, is necessary for cellular proliferation and growth, and PGE_2_ increases M2 macrophage polarization and enhances the development of colorectal cancer [24]. Therefore, we observed PGE_2_ as a primary mediator of macrophage proliferation and differentiation in the tumor microenvironment. In addition, PGE_2_ induces cancer stem cells by directly targeting colorectal neoplasia and increases the mouse oncogenic stem cell population via the NF-κB-MAPK pathway [21]. In addition to being induced in tumor cells, PGE_2_ also induces CD206-positive cells in macrophages, representing an M2 macrophage marker in the tumor microenvironment [22]. Furthermore, PMA-induced PGE_2_ was attenuated by metformin treatment. These findings suggest that decreased levels of PGE_2_ with metformin affected cancer cells and the tumor microenvironment, including tumor-promoting immune cells such as MDSCs and M2 macrophages.

To further investigate the related pathways, we evaluated the relationship between protein prenylation via the mevalonate pathway. The final steps of the mevalonate pathway include cholesterol synthesis by squalene synthase, protein geranyl-geranylation by geranylgeranyl transferase, and protein farnesylation by farnesyl transferase. Since we have already demonstrated the effect of metformin on cancer stem cells via the inhibition of protein prenylation in the mevalonate pathway in colon cancer cells [15], we focused on protein prenylation, including geranyl-geranylation and farnesylation, in macrophages of the tumor microenvironment. Protein prenylation allows the farnesylation of Ras-family proteins, while most Rho-family proteins are geranyl-geranylated. This action leads to the activation of COX-2 and PGE_2_ by inducing Erk 1/2 or Akt/NF-κB [25,26].

Previous reports have also shown that mevalonate pathway inhibitors may stimulate immune surveillance—that is, the intrinsic potential of the immune system to control or eliminate cancer [26]. In addition, inhibition of the mevalonate pathway in dendritic cells may lead to the activation of antigen-specific T-cells and NK cells, which can collaborate to produce large amounts of IFN and exhibit potent anti-tumor cytotoxicity [27,28]. Our results reveal that metformin and rapamycin reduced HMGCR induced by PMA treatment. Furthermore, the inhibition of protein prenylation by combining a protein geranylgeranyl transferase inhibitor and a farnesylation inhibitor significantly reduced the MDSC/M2 macrophage fraction. However, it was difficult to identify a change in the shifted prenylated protein band of RAS in the present study. We need to elucidate the direct relationship between the effect of metformin and the inhibition of protein prenylation in the future research. In conclusion, metformin is a potential adjunctive drug for cancer prevention and treatment via the suppression of both cancer stem cells and pro-tumor macrophages of the tumor microenvironment through the inhibition of the mevalonate pathway. Moreover, it would be interesting to identify the potential mechanistic network of metformin in both cancer stem cells and the inflammatory microenvironment in future research.

## 5. Conclusions

We investigated the mechanisms of the metformin-induced tumor suppressive effect in the field of tumor microenvironment, and demonstrated that the metformin-induced activation of AMPK and subsequent mTOR inhibition decreased the MDSC and M2 macrophage populations by downregulating the mevalonate pathway. Metformin may be a valuable adjunctive drug for potential CRC prevention and therapeutic strategy targeting the immune cells of the tumor microenvironment and tumor cells.

## Figures and Tables

**Figure 1 cancers-14-02881-f001:**
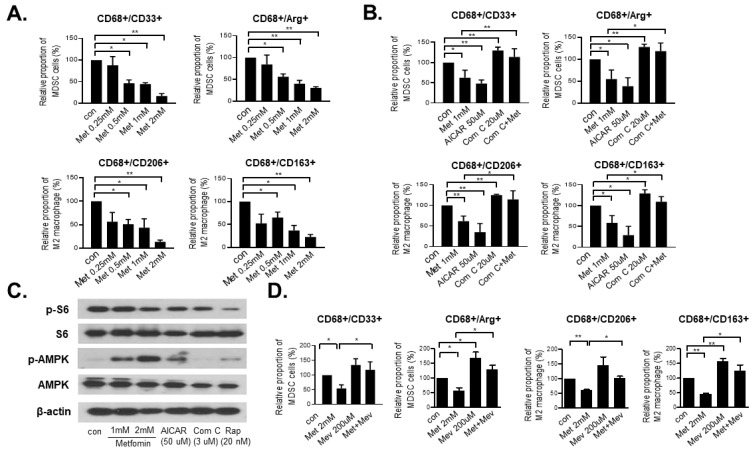
Metformin, an AMPK regulator, and mevalonate modulate the fractions of MDSCs and M1/M2 macrophages. (**A**) THP-1 cells were treated with 0.25 to 2 mM metformin in a dose-dependent manner for 48 h., and (**B**) THP-1 cells were treated with 1 mM metformin, 50 μM AICAR, 20 μM Comp C and a combination of Comp C and metformin for 48 h. Next, flow cytometry analyses were performed using the markers of MDSCs (CD68^+^ CD33^+^ and CD68^+^ Arg-1^+^), and M2 macrophages (CD68^+^ CD206^+^ and CD68^+^ CD163^+^). Data are expressed as the means ± standard errors of the four different experiments; * *p* < 0.05, ** *p* < 0.005. (**C**) In Western blot analysis, the expressions of phosphorylated S6 and phosphorylated AMPK were analyzed after 48 h of treatment with the control vehicle or 1 mM and 2 mM metformin, 50 μM AICAR, 3 μM Comp C and 20 nM rapamycin in THP-1 cells. (**D**) THP-1 cells were treated with 2 mM metformin with or without 200 μM mevalonate. Next, the expression levels of MDSC (CD33^+^/CD68^+^ and Arg-1^+^/CD68^+^) and M2 macrophages (CD206^+^/CD68^+^ and CD163^+^/CD68^+^) markers were analyzed by flow cytometry. Data are expressed as means ± standard errors of the three different experiments; * *p* < 0.05, ** *p* < 0.005.

**Figure 2 cancers-14-02881-f002:**
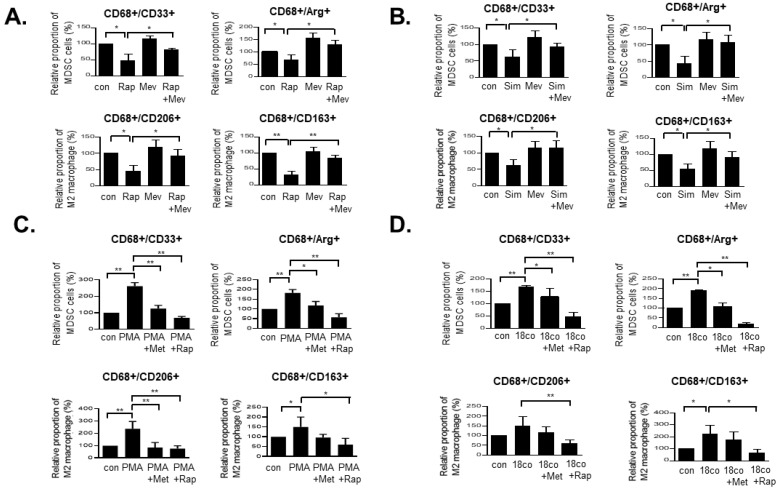
Rapamycin and simvastatin decrease the fractions of MDSCs and M2 macrophages, an effect that is reversed by mevalonate treatment. (**A**) Rapamycin (20 nM) and (**B**) simvastatin (2 μM) with or without 200 μM mevalonate were used to treat THP-1 cells for 48 h. Next, the expression levels of MDSC (CD33^+^/CD68^+^ and Arg-1^+^/CD68^+^) and M2 macrophage (CD206^+^/CD68^+^ and CD163^+^/CD68^+^) markers were analyzed by flow cytometry. Data are expressed as the means ± standard errors of the three different experiments; * *p* < 0.05, ** *p* < 0.005. THP-1 cells were pre-treated with 100 nM PMA for 6 h. (**C**) and then were treated with or without 2 mM metformin or 20 nM rapamycin for 48 h. 18Co Cells were grown in serum-supplemented culture media, washed with PBS and starved overnight in serum-free media. After 2 days, this medium was harvested, and THP-1 cells were grown in 18Co CM and treated with or without 2 mM metformin or 20 nM rapamycin for 48 h. (**D**) Next, the expression levels of MDSC (CD33^+^/CD68^+^ and Arg-1^+^/CD68^+^) and M2 macrophage (CD206^+^/CD68^+^ and CD163^+^/CD68^+^) markers were analyzed by flow cytometry analysis. Data are expressed as the means ± standard errors of the three different experiments; * *p* < 0.05, ** *p* < 0.005.

**Figure 3 cancers-14-02881-f003:**
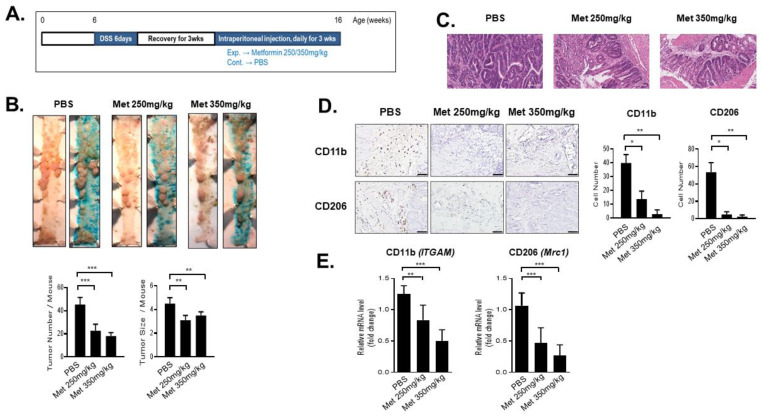
Effect of metformin on tumors of the mouse colon cancer model and immunohistochemistry analysis of MDSCs and M1/M2 macrophages. (**A**) In vivo tumorigenesis model experimental schedule, using Apc^Min/+^ treated with 3% DSS, and metformin treatment schedule. (**B**) After spraying methylene blue, the number and size of polyps in the colon were measured. Data are expressed as the means ± standard errors; ** *p* < 0.005, *** *p* < 0.001 (n = 3 per group). (**C**) Representative H&E staining of the rectum of mice (scale bar = 50 μm). (**D**) Immunohistochemistry (IHC) of the colon sections was performed on paraffin-embedded sections using MDSC (CD11b) and M2 macrophage (CD206) markers (scale bar = 50 μm). CD11b- and CD206-stained cells were counted in ten different fields under 200× magnification. Data are expressed as the means ± standard errors; * *p* < 0.05, ** *p* < 0.005 (n = 3 per group). (**E**) Mice colon tissues were homogenized and examined by real-time qPCR for CD11b (ITGAM) and CD206 (Mrc1), and normalized to β-actin expression. Data are expressed as the means ± standard errors of the three different experiments; ** *p* < 0.005, *** *p* < 0.001. n = 3 per group.

**Figure 4 cancers-14-02881-f004:**
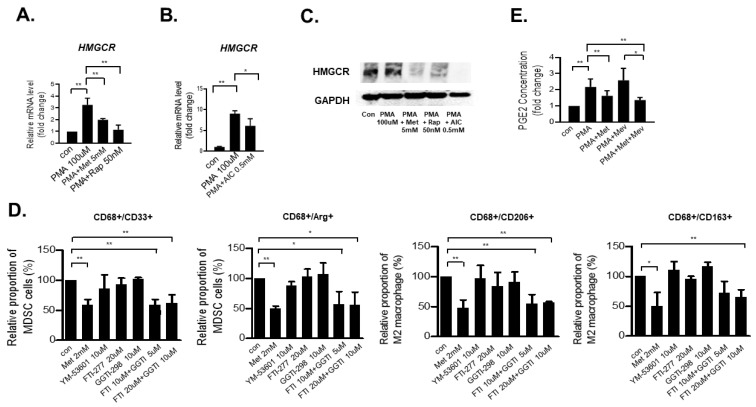
Metformin decreases the fractions of MDSCs and M2 macrophages by downregulating the mevalonate pathway for protein prenylation. (**A**) THP-1 cells were treated with PMA or control vehicle for six hours and then were treated with 5 mM metformin and 50 nM rapamycin for an additional 4 h. (**B**) Regarding the specific condition of the upstream regulator AICAR, THP-1 cells were treated with PMA and AICAR together for 4 h. Cells were harvested and examined by real-time qPCR, normalized to GAPDH expression. Data are expressed as the means ± standard errors of the two different experiments; * *p* < 0.05, ** *p* < 0.005. (**C**) In Western blot analysis, the expression of HMGCR was analyzed after 24 h of treatment with the control vehicle or PMA (100 μM) with or without metformin (5 mM), rapamycin (50 nM) and AICAR (0.5 mM) in THP-1 cells. (**D**) THP-1 cells were treated with 2 mM metformin, 10 μM YM-53601, 10 to 20 μM FTI-277 and 5 to 10 μM GGTI-298 for 48 h. Next, the expression levels of MDSC (CD33^+^/CD68^+^ and Arg-1^+^/CD68^+^) and M2 macrophage (CD206^+^/CD68^+^ and CD163^+^/CD68^+^) markers were analyzed by flow cytometry. Data are expressed as the means ± standard errors of the three different experiments; * *p* < 0.05, ** *p* < 0.005. (**E**) THP-1 cells were treated with PMA for 48 h with the control vehicle or co-treated with 2 mM metformin and/or 200 μM mevalonate. After 48 h, the culture supernatant was analyzed by ELISA. Date are expressed as the means ± standard errors of the four different experiments; * *p* < 0.05, ** *p* < 0.005.

**Figure 5 cancers-14-02881-f005:**
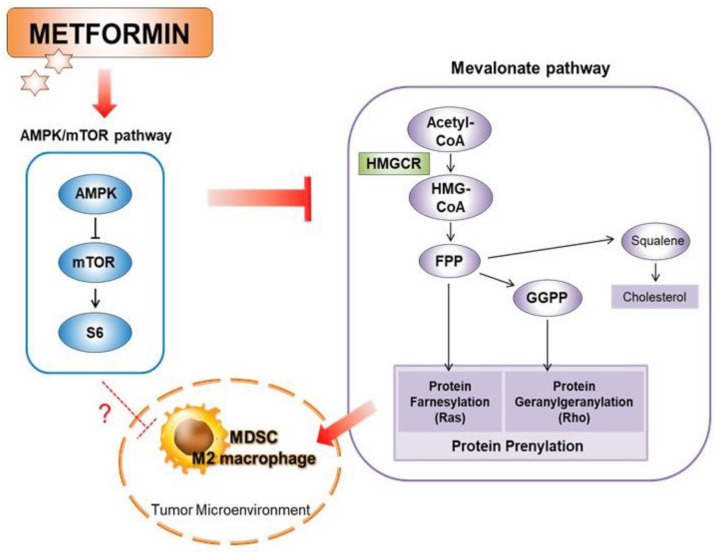
Overview of the inhibitory effect of metformin on MDSCs and M2 macrophages in the tumor microenvironment.

## Data Availability

No new data were created or analyzed in this study. Data sharing is not applicable to this article.

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
