# Peer review of "Tumor-Suppressive Effect of Metformin via the Regulation of M2 Macrophages and Myeloid-Derived Suppressor Cells in the Tumor Microenvironment of Colorectal Cancer"

_cancers, 2022, doi:10.3390/cancers14122881_

Round 1
Reviewer 1 Report
The authors have not addressed the major concern regarding the THP1 differentiation cultures. I understand that THP1 cells are difficult and very sensitive to culture conditions, but 10 to 70% variation between independent differentiation experiments is not acceptable working with a cell line. Even though the authors claim that drug effects are consistent, basal conditions are so variable that drug effect cannot be trusted. Culture conditions should be adjusted (cell numbers, passages, serum, etc…) to reduce variability, or, as I suggested in my previous report, use differentiation conditions to force differentiation (ie, IL-4, PMA). This could be a way to reduce the variability in the spontaneous differentiation of the THP1 cell line.
Author Response
Thank you for your comment. As you recommended, to reduce variation, we adjusted culture condition strictly, and performed several flow cytometry analyses again. As a result, wide variation of specific populations of THP1 cells decreased (Fig. S1 A, Fig S2 C), and we replaced Fig 1 and Fig 2 with new results from these raw data.
Reviewer 2 Report
The authors have addressed the technical concerns and somewhat improved English. However, some sentence structures remain awkward.
Author Response
Thank you for your comment. To improve the English, a native speaker reviewed this manuscript again, and some words and sentences were corrected.
Reviewer 3 Report
Line 37: name the respective inhibitors
Line 61: It is misleading that M1 phenotype inhibits tumor progression. Comment: Generally, inflammation is a critical component of tumor progression.
Line 64-65: "M2 macrophages promote and produce high levels of cytokines that stimulate tumor growth and progression with the production of antiinflammatory stimuli such as TGF-β1 and IL-10." Reference (4) does not contain any information about TGF-β1 and IL-10.
Comment:" M1 macrophages are pro-inflammatory and responsible for inflammatory signaling, while M2 is anti-inflammatory macrophages that participate in the resolution of the inflammatory process, M2 macrophages produce anti-inflammatory cytokines, thereby contributing to tissue healing, doi: 10.18632/oncotarget.24788.
Line 67: „Some of the molecules produced by M2 macrophages attract additional pro-inflammatory mediators to the tumor site, amplifying the inflammatory microenvironment.” Ref :5,
The reviewer did not find adequate information in reference 5 regarding the sentence.
Line 261: "18Co cells (intestinal 261 myofibroblasts)" Giving no information on why it was needed for the experiment, and where it came from.
Line:404: " To further investigate the related pathways, we evaluated the relationship between protein prenylation via the mevalonate pathway"
Comment: Such data is not included in the MS, and the reference (14) as a reference for prenylation is not very convincing.
Ras prenylation suppression is one of the main claims in MS, the authors should provide a well-separated RAS / prenylated-RAS WB showing the entire blot for review from 3 independent samples at least.
Both the introduction and the discussion are not unequivocal at all, confusing in their present form.
Author Response
Line 37: name the respective inhibitors
- Response: Thank you for your kind comment. We added each inhibitor in indicated sentence (line 39-40).
Line 61: It is misleading that M1 phenotype inhibits tumor progression. Comment: Generally, inflammation is a critical component of tumor progression.
- Response: Thank you for your comment. As you know, the inflammation of tumor microenvironment contributes to both suppression and progression of tumor according to the characteristics of inflammatory cells (Ostrand-Rosenberg, S. Immune surveillance: a balance between protumor and antitumor immunity. Curr Opin Genet Dev 2008, 18, 11-18, DOI:10.1016/j.gde.2007.12.007.). M1 macrophage is one of the inflammatory cells related to tumor rejection.
Line 64-65: "M2 macrophages promote and produce high levels of cytokines that stimulate tumor growth and progression with the production of antiinflammatory stimuli such as TGF-β1 and IL-10." Reference (4) does not contain any information about TGF-β1 and IL-10.
Comment:" M1 macrophages are pro-inflammatory and responsible for inflammatory signaling, while M2 is anti-inflammatory macrophages that participate in the resolution of the inflammatory process, M2 macrophages produce anti-inflammatory cytokines, thereby contributing to tissue healing, doi: 10.18632/oncotarget.24788.
- Response: Thank you for your comment. As you mentioned, the reference (4) is not enough for this description. As you know, the interaction of inflammatory cells for tumor suppression and progression is complicated, and the inflammatory conditions of general inflammation and tumor microenvironment are different. Therefore, we added a reference related with pro-tumoral and anti-tumoral inflammatory signal in tumor microenvironment (line 67, Ref 26; Ostrand-Rosenberg, S. Immune surveillance: a balance between protumor and antitumor immunity. Curr Opin Genet Dev 2008, 18, 11-18, DOI:10.1016/j.gde.2007.12.007).
Line 67: „Some of the molecules produced by M2 macrophages attract additional pro-inflammatory mediators to the tumor site, amplifying the inflammatory microenvironment.” Ref :5,
The reviewer did not find adequate information in reference 5 regarding the sentence.
- Response: Thank you for your comment. Actually, we would like to describe the complexity and various interactions of inflammatory signaling in tumor condition, but this sentence could cause some confusion. Therefore, we removed this sentence.
Line 261: "18Co cells (intestinal 261 myofibroblasts)" Giving no information on why it was needed for the experiment, and where it came from.
- Response: Thank you for your detailed comment. We already mentioned the reason we used 18Co cells in line 260-262, “Furthermore, to show the same effect of metformin in the macrophage-activated state, we induced macrophage activation using PMA and conditional media (CM) of 18Co cells (intestinal myofibroblasts), critical cell components of the tumor microenvironment.” As you know, in tumor condition, infiltrated myofibroblasts have a critical role in tumor progression, like macrophage activation. We published a related paper “Kim, J. H., et al. (2012). The role of myofibroblasts in upregulation of S100A8 and S100A9 and the differentiation of myeloid cells in the colorectal cancer microenvironment. Biochemical and Biophysical Research Communications 423(1): 60-66.”, and added this reference (13) in the sentence (line 262).
Line:404: " To further investigate the related pathways, we evaluated the relationship between protein prenylation via the mevalonate pathway"
Comment: Such data is not included in the MS, and the reference (14) as a reference for prenylation is not very convincing.
Ras prenylation suppression is one of the main claims in MS, the authors should provide a well-separated RAS / prenylated-RAS WB showing the entire blot for review from 3 independent samples at least.
- Response: Thank you for your comments. As you recommended, we performed Western blot analysis to identify a change of shifted prenylated protein band of RAS. However, despite decreased expression of RAS by metformin treatment, we could not find a definite shifted prenylated protein band of RAS (Fig.S7). Therefore, we added the following sentences in Results (line 333-336, “However, in Western blot analysis performed to identify a change of shifted prenylated protein band of RAS, despite decreased expression of RAS by metformin treatment, we could not find any definite shifted prenylated protein band of RAS (Fig. S7).”) and Discussion (line 421-424, “However, it was difficult to identify a change of shifted prenylated protein band of RAS in the present study. We need to elucidate the direct relationship between the effect of metformin and the inhibition of protein prenylation in the future research.”), and removed words or sentences related with metformin-induced protein prenylation in abstract, results, discussion and conclusion.
This manuscript is a resubmission of an earlier submission. The following is a list of the peer review reports and author responses from that submission.
Round 1
Reviewer 1 Report
The manuscript by Kang et al proposes that the tumour suppressor activity of metformin in colorectal cancer is mediated by the reduction in the differentiation of myeloid-derived suppressor cells (MDSC) and M2 macrophages. Mechanistinally, the authors propose that metformin inhibits the mevalonate pathway-induced protein prenylation. These studies are mostly based in in vitro differentiation assays of the human monocyte cell line THP-1 upon PMA stimulation in presence of different drugs, combined with an in vivo model of colorectal cancer (DSS on APC+/- mice) with metformin treatment.
The THP1 differentiation assays presented in the manuscript are a major concern for this reviewer. According to the authors, PMA stimulation induces the differentiation of THP1 cell line into three different populations that are identified with different surface markers: MDSC (CD68+CD33+ and CD68+Arg+), M2 macrophages (CD68+CD206+ and CD68+CD163+ and M1 macrophages (CD68+iNOS+). However, all the representative dot plots in the supplementary figures show uniform population with a diagonal shape when CD68 is plotted against any of the other markers. If the diagonal were real, it would mean that all cells express all markers, maybe to a different extend, but there are no different populations within the culture with the selected markers. For example, in fig 1b, according to dot plots, there are 74% MSDC and 73% M2, that is not possible if they are different populations. I believe there is a major problem of flow cytometry analysis in the samples, as diagonal shapes in flow cytometry often indicate compensation problems. The authors should check compensations, and if the diagonals are really true, consider changing the selected markers to identify the different populations, because with the data as it is I cannot distinguish three different populations within the culture. Most of the conclusions in the study are supported by the THP1 differentiation assays, but the poor quality of the presented FACS plots does not sustain the authors claims.
The authors should include additional controls:
- THP1 without PMA. This is the only way to show that PMA is inducing differentiation into the three different lineages.
- All graphs display the relative proportion of a frequency, and making a percentage out of a data that already is a percentage is confusing and may be misleading. As the FACS data is already a frequency, graphs should display the real percentage extracted from FACS dot plots, it doesn´t need to be normalised. This way it is also possible to assess how efficient are the differentiation cultures in the generation of the three different populations.
- There is an extremely high variability in the differentiation assays in the supplementary figures. For example, in one experiment there are 10% of MSDC in the control condition, and more than 70% in another. It maybe a problem of the FACS analysis, because with that level of variability within cultures it is very hard to extract conclusions.
- Include a cell viability marker to exclude dead cells in the stainings. In the supplementary figures (ie, S1A), there are small diagonals that clearly look like dead cells that are currently being considered as double positive cells.
- Consider the differentiation assays in presence of IL-4 to force the differentiation towards M2 macrophages, the metformin effect must be stronger in those conditions.
- Use primary cells to validate some of the key experiments.
Finally, I would like to mention that none of the experiments directly show that the metformin effects are mediated by AMPK activation. Metformin activates AMPK in an indirect manner, by altering the ATP/AMP ratio in the cell. Thus, metformin activates many other ATP/AMP-sensitive enzymes. In fact, the anti-diabetic effects of the metformin are largely independent of AMPK activation. The authors should consider alternative strategies to provide evidence of a direct link between AMPK activation and observed effects (ie, use AMPK direct activators such as A-769662, combine drugs with genetic approaches such as shRNA/CRISPR for AMPK to determine if drug effects are lost).
Reviewer 2 Report
The manuscript by Kang J. et al. examines the impact of metformin on the regulation of MDSC and M2 macrophages in the colon tumor microenvironment. The study demonstrates in vitro that the metformin reduces the number of MDSC and M2 macrophages via downregulation of mevalonate pathway-induced protein prenylation. Although the beneficial effects of metformin on CRC have been suggested, the mechanisms have not been clearly understood. Therefore, the rationale for the study is reasonable, and the experiments are well-designed to address the multi-faceted effects of metformin in CRC. However, the authors have previously shown the effects of metformin on AMPK-mevalonate pathway axis, and the current study was done based on the same concept with different cell line. Use of the ApcMin model was interesting but the manuscript may be improved if the potential mechanistic network of metformin on both cancer stem cells (previous publication) and MDSCs (current study) was investigated further in this in vivo model.
Author Response
- The manuscript by Kang J. et al. examines the impact of metformin on the regulation of MDSC and M2 macrophages in the colon tumor microenvironment. The study demonstrates in vitro that the metformin reduces the number of MDSC and M2 macrophages via downregulation of mevalonate pathway-induced protein prenylation. Although the beneficial effects of metformin on CRC have been suggested, the mechanisms have not been clearly understood. Therefore, the rationale for the study is reasonable, and the experiments are well-designed to address the multi-faceted effects of metformin in CRC. However, the authors have previously shown the effects of metformin on AMPK-mevalonate pathway axis, and the current study was done based on the same concept with different cell line. Use of the ApcMin model was interesting but the manuscript may be improved if the potential mechanistic network of metformin on both cancer stem cells (previous publication) and MDSCs (current study) was investigated further in this in vivo model.
-->
Thank you for your keen comments.
As you mentioned, we previously reported the same pathway in CSC(cancer stem cell)-suppressing effect of metformin. Actually, we found the metformin-induced MDSC/M2 regulation earlier than previous report on CSC-supressing effect. After identification of mevalonate pathway-induced protein prenylation mechanism in CSC-supressing effect of metformin, we applied this pathway to the metformin-induced MDSC/M2 regulation.
We think that if we were to combine the CSC contents with inflammatroy microenvironment, it would be very complicated. Therefore, we would like to focus on inflammatroy microenvironment in these results.
However, we added the comment about the potential mechanistic network of metformin on both cancer stem cells and inflammatroy microenvironment in the discussion section (Line 427-429).
“Moreover, it would be interesting to identify the potential mechanistic network of metformin on both cancer stem cells and inflammatroy microenvironment in the future study.”
Reviewer 3 Report
This manuscript extends to myeloid cells previous studies by the authors that identified protein prenylation as an AMPK-dependent target for metformin activity in cancer stem cells. The experiments are logically designed and generally support the conclusions drawn. However, given the complexity of metformin actions, some alternative mechanisms need to be evaluated.
- The data in Fig 1D, Fig. 2, and Fig 4 are consistent with the proposed mechanism, but the authors need to consider another published mechanism for metformin to regulate cholesterol homeostasis in THP1 cells and in vivo (PMID: 29499335). Those authors concluded that metformin also regulates cholesterol levels via controlling its uptake mediated by SREBP2. Does the present data exclude an alternative mechanism?
- Interpretation of the metformin effector mechanisms in vivo in Fig. 3 is complicated by its ability to act in a cell autonomous manner on any given cell type while simultaneously initiating indirect effects on the same cell by altering the microbiome and levels of local and systemic metabolites produced by those microbes (e.g. PMID: 34158423, PMID: 33808194, PMID: 33603734, PMID: 34264978, PMID: 35115909). A recent study of metformin effects on macrophages highlights this complexity (PMID: 34264978). Metformin induced changes in the gut microbiome that altered immunoregulatory short chain fatty acid metabolite levels. Could these or other altered microbial metabolites contribute to the changes in the intestinal microenvironment observed in the present manuscript?
- English usage and grammar are awkward in places. The manuscript needs editing by a native speaker.
Author Response
This manuscript extends to myeloid cells previous studies by the authors that identified protein prenylation as an AMPK-dependent target for metformin activity in cancer stem cells. The experiments are logically designed and generally support the conclusions drawn. However, given the complexity of metformin actions, some alternative mechanisms need to be evaluated.
- The data in Fig 1D, Fig. 2, and Fig 4 are consistent with the proposed mechanism, but the authors need to consider another published mechanism for metformin to regulate cholesterol homeostasis in THP1 cells and in vivo (PMID: 29499335). Those authors concluded that metformin also regulates cholesterol levels via controlling its uptake mediated by SREBP2. Does the present data exclude an alternative mechanism?
--> Thank you for your comment. As for cholesterol synthesis pathway in mevalonate pathway, we tested it by using YM-53601 (squalene synthase inhibitor) which inhibits cholesterol synthesis in Fig 4C, but there was no significant change by YM-53601 treatment (Fig 4C). From this result, we thought that the main related pathway would be protein prenylation pathway (Fig 5).
- Interpretation of the metformin effector mechanisms in vivo in Fig. 3 is complicated by its ability to act in a cell autonomous manner on any given cell type while simultaneously initiating indirect effects on the same cell by altering the microbiome and levels of local and systemic metabolites produced by those microbes (e.g. PMID: 34158423, PMID: 33808194, PMID: 33603734, PMID: 34264978, PMID: 35115909). A recent study of metformin effects on macrophages highlights this complexity (PMID: 34264978). Metformin induced changes in the gut microbiome that altered immunoregulatory short chain fatty acid metabolite levels. Could these or other altered microbial metabolites contribute to the changes in the intestinal microenvironment observed in the present manuscript?
--> Thank you for your kind information. I fully agree with your opinion, but comments on microbiome factor would be out of scope of our results.
- English usage and grammar are awkward in places. The manuscript needs editing by a native speaker.
--> We rechecked English by a native speaker.
Reviewer 4 Report
The MS presents the effect of metformin on M2 macrophages and MDSCs using monocyte THP-1 cells and a dextran sodium sulfate treated ApcMin/+ mouse model of colon cancer with the conclusion that activation of AMPK and subsequent mTOR inhibition decreased the MDSC and M2 macrophage populations by downregulation of protein prenylation through mevalonate pathway suppression.
Major recommendations and Comments:
In a few sentences, Arthurs should introduce why they chose CD33, arginase, CD206, and CD136 markers.
Check the meaning of this sentence: “Furthermore, regarding the anti-tumor effect, metformin targets the respiratory chain complex I in the mitochondria, leading to metabolism, including the TCA (tricarboxylic acid) cycle and oxidative phosphorylation (OXPHOS)”
LINE 112-113: How long the PMA treatment lasted, regarding the M2 type Macrophage polarization.
LINE 122: How long did the BSA containing pretreatment last?
Line 135: “Cells were treated with 1 mM, 2.5 mM or 5 mM metformin”
Cells culture cannot be treated with concentration, Reviewer assumes this was the concentration of metformin in the cell culture.
Line: 162 “according to institutional guidelines and policies”. The ID of the animal experimental license is required.
Calculation of the Relative proportion of MDSC cells (%) is not described.
LINE 316: “Thus, we analyzed the expression level of HMGCR after metformin treatment or mTOR inhibition.” This data is not shown, it would be helpful to show them by qPCR and Western blot.
LINE 318-319: “HMGCR expression in THP-1 cells was increased when treated with PMA only, whereas PMA-induced HMGCR expression was attenuated by treatment with metformin, rapamycin (Fig. 4A). Show the protein levels of HMGCR in each case by Western blot (Fig 4A, B.)
LINE 364 -367: The statement is just an assumption, Reviewer didn't find any data (Western blots) for that to be proven.
LINE 394-394: “and PGE2, an end product of the mevalonate pathway,” This is a false statement.
- The figures' quality is not fine; if the reader tries to magnify it, the titles of the axes are not visible, which is typical of all of the figures' labeling.
- As the authors have stated a few times that: e.g. LINE 382: suggesting that the AMPK/mTOR pathway and mevalonate pathway could be key mediators of MDSC/M2 macrophage regulation- A protein validation of AKT or mTOR by Western blots would support that statement further, without that one, this is a little bit speculative.
- The authors should evaluate their results with further Western blot and qPCR (CD11b CD206, PGE2) measurements.
Minor recommendations
- The figures have a strong border of the graphs/bars which is by a magnification gets worst would be beneficial to change to 1pt or 1.5 pt even on the SD values.
- The figures in the supplement – flow cytometry is the weakest point of the data set; unable to read it, the magnification does not help.
Reviewer 5 Report
Reviewer comments:
Comments to the Author
This manuscript describes the inhibitory effect of metformin on MDSCs and M2 macrophages in the tumor microenvironment of colon cancers is mediated by AMPK activation and subsequent mTOR inhibition, leading to downregulation of protein prenylation through mevalonate pathway suppression. In dextran sodium sulfate treated ApcMin/+ mouse colon cancer model, metformin reduced the number and volume of colorectal tumors with decreased populations of Myeloid-derived suppressor cells (MDSCs) and M2 macrophages in the tumor microenvironment.
The experimental designing is impressive, and the manuscript is for the most part well written with substantial evidence of confirmatory and supplementary data. The discussion is also well goes with the results and postulated according to the evidence provided. The references are appropriate and timely.
Minor criticisms
• In every figure legend the information about number of samples used to conduct each experiment is not provided. Authors are advised to provide N values for each experiment in the figure legends.
• Please undergo a thorough check of the manuscript for typographical and grammatical errors.
Author Response
Comments to the Author
This manuscript describes the inhibitory effect of metformin on MDSCs and M2 macrophages in the tumor microenvironment of colon cancers is mediated by AMPK activation and subsequent mTOR inhibition, leading to downregulation of protein prenylation through mevalonate pathway suppression. In dextran sodium sulfate treated ApcMin/+ mouse colon cancer model, metformin reduced the number and volume of colorectal tumors with decreased populations of Myeloid-derived suppressor cells (MDSCs) and M2 macrophages in the tumor microenvironment.
The experimental designing is impressive, and the manuscript is for the most part well written with substantial evidence of confirmatory and supplementary data. The discussion is also well goes with the results and postulated according to the evidence provided. The references are appropriate and timely.
Minor criticisms
• In every figure legend the information about number of samples used to conduct each experiment is not provided. Authors are advised to provide N values for each experiment in the figure legends.
• Please undergo a thorough check of the manuscript for typographical and grammatical errors.
--> Thank you for kind comments. We added N values in figure legends, and rechecked the English.